# Adolescent Health and Dark Personalities: The Role of Socioeconomic Status, Sports, and Cyber Experiences

**DOI:** 10.3390/ijerph21080987

**Published:** 2024-07-27

**Authors:** Ilrang Lee, Yonghwan Chang, Ye Lei, Taewoong Yoo

**Affiliations:** 1School of Special Education, School Psychology, and Early Childhood Studies, College of Education, University of Florida, Gainesville, FL 32611, USA; ilranglee@coe.ufl.edu; 2Department of Sport Management, College of Health and Human Performance, University of Florida, Gainesville, FL 32611, USA; yelei@ufl.edu (Y.L.); taewoong.yoo@ufl.edu (T.Y.)

**Keywords:** dark personality, adolescent health, socioeconomic factor, sports engagement, cyber victimization

## Abstract

By investigating the impact of dark personality traits on adolescent health, this study explores the interplay among economic disadvantage, spectator sports involvement, and cyber victimization. We analyzed data from 1844 students aged 13–17 in a school district in the USA, and our findings reveal positive associations between economic disadvantage and both Machiavellianism and psychopathy, influencing negative emotions-driven eating. Spectator sports engagement exhibits links with Machiavellianism and narcissism, suggesting potential drawbacks to competitive behaviors. Cyber victimization shows associations with psychopathy and negative emotions-driven eating. The results illuminate the dynamic associations between emotional eating and health outcomes, including BMI and perceived quality of life. These findings deepen our understanding of how dark personality is shaped and subsequently influences adolescents’ well-being, offering insights for targeted interventions.

## 1. Introduction

Adolescence, characterized by rapid physical, psychological, and social changes, significantly shapes lifelong well-being [1] Numerous studies [2,3] have explored factors influencing these transformations. However, predominant research often emphasizes the positive aspects of this developmental stage [4,5], leaving a significant gap in understanding the potential impact of dark personalities—traits including manipulation (Machiavellianism), self-centeredness (narcissism), and a lack of empathy (psychopathy) [6]. Despite widespread discussions on dark personalities in social and developmental psychology and their potential consequences in various life domains [4], their specific influence on adolescent health remains largely unexplored [7,8]. Investigating dark personalities during adolescence is theoretically justified given the potential long-term consequences associated with these traits [9]. These traits, for example, can hinder social relationships and academic success [10], while also contributing to negative coping mechanisms, such as emotional eating, ultimately affecting physical health and overall quality of life [4].

To address these complexities, we chose three key determinants for our study: economic disadvantage, involvement in spectator sports, and experiences of cyber victimization. Economic disadvantage is included due to research suggesting a potential link between socioeconomic factors such as financial stressors and the manifestation of dispositional traits and persistent behavior (i.e., social cognitive adaptations and evolutionary psychology approach) [11,12,13]. Furthermore, our decision to investigate the involvement in spectator sports as a determinant is grounded in emerging evidence indicating an association among viewership activities, sports engagement, and the development of certain personality traits, such as aggression and violent behavior, in adolescence (i.e., competitive mindset principle) [14,15,16]. The inclusion of experiences of cyber victimization as contemporary challenges is based on the growing recognition of the profound influence of online interactions on adolescent development in the digital age (i.e., digital self-defense hypothesis) [17,18,19].

Taken together, our research endeavors to address critical gaps in understanding with the following objectives: First, we aim to uncover the manifestations of dark personalities in adolescents, providing an improved understanding of how these traits manifest during this critical developmental phase. Second, we seek to identify key determinants shaping the development of dark personalities, exploring potential links with economic disadvantage, involvement in spectator sports, and experiences of cyber victimization. Last, we intend to explore the complex relationships among these key determinants, dark personalities, emotional eating (both positive and negative), and their subsequent effects on body mass index (BMI) and health-related quality of life (HRQoL). The ultimate goal of our research is to safeguard adolescent psychophysical health, promote emotional well-being, and establish a foundation for their thriving future.

## 2. Theoretical Background and Hypothesis Development

### 2.1. Impact of Adolescent Dark Personalities on Well-Being

Dark personalities are characterized by traits such as manipulation, a grandiose sense of self, and a lack of empathy [20]. These traits, often associated with negative social behaviors [10], have been extensively studied in various contexts; however, their specific implications for adolescent health remain less explored. Originally stemming from social psychology scholars [20] and then psychometrically validated by a number of follow-up studies [5], dark personalities are commonly defined by the following three unique traits: Machiavellianism, narcissism, and psychopathy. First, rooted in the philosophy of Niccolò Machiavelli, Machiavellianism reflects a strategic and manipulative approach to interpersonal relationships [9]. In adolescents, Machiavellian traits may manifest as cunning strategies, deceitful maneuvers, and a willingness to exploit others [9]. Research suggests that these traits can significantly influence social dynamics [10], potentially impacting the psychophysiological and social well-being of adolescents [20].

Second, narcissism involves an excessive focus on oneself, coupled with a constant need for admiration and a lack of empathy [20]. In adolescents, narcissistic traits may contribute to self-centered behaviors, an inflated self-image, and distinct interpersonal dynamics [9]. Research suggests that behavioral tendencies associated with narcissism could significantly shape adolescent health [21]. Therefore, it is imperative to closely examine how these self-focused tendencies impact relationships, emotional well-being, and the development of a healthy self-concept during the formative years [20]. Third, psychopathy encompasses traits such as impulsivity, a lack of remorse, and a diminished capacity for empathy [20]. While traditionally associated with criminal behavior [10], there is an increasing suggestion that its manifestation in adolescence can have broader implications for social relationships and emotional well-being [22].

### 2.2. Determinants of Adolescent Dark Personalities

#### 2.2.1. Economic Disadvantage

Economic disadvantage, characterized by constrained financial resources and limited opportunities, is posited to exert a profound influence on the psychological development of adolescents through the lens of social cognitive adaptations and evolutionary psychology [11,12,13] In this challenging socioeconomic landscape, adolescents may be driven to adopt specific dark personality traits as adaptive strategies, shaped by evolutionary pressures for survival and resource acquisition [23,24]. The limited availability of financial resources can evoke feelings of powerlessness [25], prompting the need for strategic thinking to secure essential resources and opportunities [22]. Machiavellianism, characterized by a strategic and manipulative approach to social interactions [11], may manifest as a pragmatic response to the challenges imposed by economic adversity, aligning with evolutionary principles of adapting to resource constraints for survival [23].

Moreover, an inflated sense of self-importance may emerge as an adaptive coping mechanism [10,11] against the potential negative impact of economic adversity on self-esteem [25] This reflects the evolutionary drive for social status in resource-limited environments [22]. Simultaneously, economic deprivation may foster the emergence of psychopathic traits in adolescents as they confront stressors and adversities associated with limited financial resources. A lack of empathy, impulsivity, and a disregard for social norms may represent maladaptive responses to the chronic challenges [26] posed by economic deprivation, echoing the evolutionary imperative to overcome obstacles and secure resources [24].

**Hypothesis 1.** *Economic disadvantage is positively associated with Machiavellianism (H1a), narcissism (H1b), and psychopathy (H1c) among adolescents*.

#### 2.2.2. Spectator Sports Involvement

Active engagement in sports spectatorship, marked by the observation of sporting events and following athletes and teams, may introduce adolescents to a distinct social context. According to the competitive mindset principle [14,15,16], the competitive nature of sports, coupled with the dynamics of team engagement, plays an important role in shaping psychological characteristics. Within the unique context of sports spectatorship, where strategic gameplay, team dynamics, and competition unfold [27], adolescents may be exposed to Machiavellian tactics employed by athletes, coaches, or even fellow fans within the sports culture [28]. This exposure may contribute to the adoption of Machiavellian traits among spectators, with the competitive nature of sports serving as a model for strategic and manipulative behavior [27].

Similarly, the competitive and achievement-oriented nature of sports can foster narcissistic traits among adolescent spectators. The competitive mindset principle highlights how the competitive dynamics inherent in sports can influence individuals to exhibit traits associated with an inflated sense of self-importance [15]. Meanwhile, the intensity, aggression, and emotional dynamics [29] witnessed in sports may also contribute to the development of psychopathic traits among spectators. In other words, the desensitization to violence, coupled with the thrill-seeking nature of sports engagement [15,16], may shape the psychopathic characteristics of adolescents immersed in sports culture.

**Hypothesis 2.** *Sports spectatorship involvement is positively associated with Machiavellianism (H2a), narcissism (H2b), and psychopathy (H2c) among adolescents*.

#### 2.2.3. Cyber Victimization

Experiences of cyber victimization, stemming from online harassment and bullying in the digital age [18] are postulated to significantly influence adolescent health outcomes. According to the digital self-defense hypothesis [17,18,19], individuals, in response to online threats, consciously or subconsciously adopt psychopathic traits such as coldness and manipulation as self-defense mechanisms [30]. The manipulative nature of Machiavellianism can emerge as a means for victims to navigate online interactions, safeguard themselves [21], and gain control [17] in response to perceived threats. Similarly, experiences of online harassment may contribute to the development of narcissistic traits as victims seek to fortify their self-esteem in the face of cyber threats [19]. The pursuit of an augmented self-image may serve as adaptive coping mechanisms against the potential negative impact of cyber victimization on self-worth [31], in accordance with the principles of the digital self-defense hypothesis [18]. Moreover, the challenges posed by online harassment may lead to the development of psychopathic traits [8]. In other words, a lack of empathy, impulsivity, and a disregard for social norms may surface as maladaptive responses to the chronic stressors associated with cyber victimization [21].

**Hypothesis 3.** *Cyber victimization is positively associated with Machiavellianism (H3a), narcissism (H3b), and psychopathy (H3c) among adolescents*.

### 2.3. Dark Personalities and Emotional Eating

The exploration of dark personalities in adolescence extends to their potential influence on emotional eating behaviors. Emotional eating, characterized by the consumption of food in response to emotions rather than hunger, could be a critical aspect of adolescent health [32]. Conventionally, emotional eating largely refers to negative emotions-driven eating as a form of dissociation [33]. However, recent studies increasingly highlight the impact of favorable emotions-eliciting eating behavior [34], given its relatively less focused but potentially large impact on health outcomes [35]. The theoretical basis for investigating the impact of positive emotions-driven eating lies in recent evidence that positive affect can significantly influence dietary choices and eating patterns [32]. Recognizing the potential dual influence of emotions on eating habits, this section addresses both types of eating more specifically, assigning equal importance to each for a comprehensive examination of their role in adolescent health.

Positive emotions-driven eating involves consuming food to enhance positive emotions or reward oneself [36]. Machiavellianism, known for its strategic and manipulative approach to interpersonal relationships [10], may lead adolescents to adopt positive emotions-driven eating as a strategic coping mechanism. The calculated nature of Machiavellian individuals may drive them to use food as a tool for reward or reinforcement in response to positive emotions [36]. For example, adolescents with Machiavellian traits might strategically engage in positive emotions-driven eating, such as indulging in a favorite snack or dessert, to uplift their mood or reinforce a sense of personal reward. Similarly, the pursuit of an uplifted self-image and the desire for admiration [6] may lead narcissistic adolescents to use specific foods as a means of reinforcing their positive self-perception. For example, adolescents with narcissistic traits might selectively indulge in celebratory meals or treats, viewing them as symbolic rewards that enhance their already elevated self-image and attract admiration from their classmates. Psychopathy, quite simply, may lead adolescents to engage in positive emotions-driven eating as a means of impulsively seeking pleasure or gratification (i.e., hedonic eating; [37]).

**Hypothesis 4.** *Machiavellianism (H4a), narcissism (H4b), and psychopathy (H4c) are positively associated with positive emotions-driven eating among adolescents*.

Conversely, negative emotions-driven eating refers to the consumption of food as a coping mechanism for unpleasant emotions such as stress, sadness, or anxiety [34]. Adolescents exhibiting Machiavellian, narcissistic, or psychopathic traits may turn to negative emotions-driven eating as a means of managing their emotional states. Machiavellianism, characterized by strategic and manipulative behaviors [7], can shape adolescents’ inclination toward negative emotions-driven eating as a coping mechanism. The calculated and cunning nature of Machiavellian individuals may prompt them to employ food as a means of emotional manipulation or control, leveraging it strategically in their interpersonal dynamics. Driven by the need for social recognition and the pursuit of an impressive self-image [23] individuals with narcissistic traits may resort to using food as a tool to maintain or enhance their perceived self-worth. The act of consuming certain foods could serve as a visible expression of their elevated status and distinct self-image [36]. Psychopathy, characterized by impulsivity and a lack of empathy [25], may influence adolescents to resort to negative emotions-driven eating as a means of regulating their emotional states. The emotional detachment and impulsive behaviors associated with psychopathy [7] may lead individuals to turn to food as a quick and accessible way to modulate their mood (i.e., dissociative eating; [33]).

**Hypothesis 5.** *Machiavellianism (H5a), narcissism (H5b), and psychopathy (H5c) are positively associated with negative emotions-driven eating among adolescents*.

### 2.4. Health Implications of Emotional Eating

Emotional eating, influenced by dark personalities, could serve as a gateway to health-related consequences in adolescence. Our primary focus centers on two critical health outcomes: BMI and HRQoL. BMI, functioning as a quantitative measure of body fat derived from weight and height [38] provides an objective assessment. Meanwhile, HRQoL captures an individual’s perception of overall well-being across physical, mental, and social domains [39]. [Given this background, consuming food driven by positive emotions, often motivated by the pursuit of pleasure or gratification in response to positive affective states [32,38], may contribute to an increase in BMI among adolescents. The inclination to seek comfort or reward through eating during positive emotional states can lead to the consumption of calorie-dense foods [33,40], potentially resulting in weight gain. Relying on food as a coping mechanism for boosting emotions may lead to maladaptive eating patterns [34,36] that, over time, contribute to physical health concerns and an overall decrease in well-being perceptions [41].

**Hypothesis 6.** *Positive emotions-driven eating is positively associated with BMI (H6a) but negatively associated with HRQoL (H6b) among adolescents*.

Similarly to positive emotions-driven eating, we argue that eating driven by negative emotions, catalyzed by the desire for emotional comfort or distraction through food in response to negative emotional states, may contribute to an increase in BMI among adolescents. The tendency to seek solace or distraction through eating during negative emotions can lead to the consumption of calorie-dense foods [38], mirroring the outcomes associated with positive emotions-driven eating and potentially resulting in weight gain. Negative emotions-driven eating, while providing temporary relief from negative emotional states [36], may detrimentally impact HRQoL among adolescents. In other words, despite offering temporary emotional relief, negative emotions-driven eating may still have a harmful impact on HRQoL.

**Hypothesis 7.** *Negative emotions-driven eating is positively associated with BMI (H7a) but negatively associated with HRQoL (H7b) among adolescents*.

## 3. Methods

### 3.1. Samples and Procedures

We collaborated with an Independent School District (ISD) in the USA. The ISD, serving approximately 18,000 students from diverse demographic backgrounds, is suitable for cross-sectional survey studies. Data for this study were collected from students between 17 April and 28 April 2023, using the ISD’s online platform to complete questionnaires. Students, utilizing the computers in their individual classrooms, were allocated approximately twenty minutes during class time to complete the questionnaire. Additionally, students received survey links via their ISD email accounts, allowing them the flexibility to complete the survey at their convenience and from any location. Participation was voluntary, and no payment was involved. Prior to data collection, the institutional review board examined and approved a protocol to ensure adherence to ethical guidelines and data privacy laws. Parents or legal guardians provided consent before students could participate. The study encompassed children in the ISD’s seventh to twelfth grades, approximately 2700 enrolled students, aged between 13 and 17.

### 3.2. Measures

Economic disadvantage was assessed by using data from the ISD schools’ student directory. Economic disadvantage status was obtained from the school administrative team, categorizing students into two groups based on the existing guidelines [24,26]: those eligible for free or reduced-price meals with household incomes at or below 130% of the federal poverty level (coded as 1 for economic disadvantage) and those not meeting these criteria (coded as 0 for no economic disadvantage).

To assess spectator sports involvement, we employed a set of four items in alignment with established conceptualizations of involvement [42]. These items were designed to measure both the frequency and depth of engagement in sports activities, capturing a comprehensive view of participants’ involvement. The first item probed the frequency of sports watching. Participants were asked to indicate the frequency of their sports consumption on a scale ranging from 1 (*never*) to 7 (*almost every day*). The second item focused on social engagement in sports. Responses were captured on a 7-point scale, ranging from 1 (*never*) to 7 (*very often*). For the third item, participants were encouraged to share their preferences regarding favorite sports, teams, and athletes. The open-ended question allowed for multiple responses, enabling participants to express a diverse range of sporting interests. The final item sought to measure the level of involvement in watching favorite teams and athletes. Participants were asked to rate the depth of their engagement on a 7-point scale, ranging from 1 (*not involved at all*—“*I don’t really watch their games or competitions*”) to 7 (*super involved*—“*I never miss a game or competition!*”).

In measuring cyber victimization, participants engaged with three distinct statements designed to capture their experiences in online and digital interactions. These statements were adapted from earlier studies [43] and adjusted to suit the specific context of the present study. In assessing the Dark Triad personality traits—narcissism, Machiavellianism, and psychopathy—we employed the 27-item Short Dark Triad developed by Jones and Paulhus [5]. This instrument consists of nine items dedicated to each of the three components, capturing the distinctive facets of these traits. Participants were instructed to respond to all items by using a Likert-type scale, where 1 denoted *strongly disagree*, and 7 represented *strongly agree*.

We adapted existing scale items to assess negative emotions-driven eating, incorporating the emotional eating scales adapted for children and adolescents [32]. When tailoring the scales to adolescents aged 13–17, we selected five emotions pertinent to the unique challenges of this age group—sadness, nervousness, anger, guilt, and not doing enough. Participants were instructed to evaluate how these emotions influenced their inclination to eat, utilizing a 7-point Likert scale ranging from 1 (*no desire to eat*) to 7 (*very strong desire to eat*). We developed a positive emotions-driven eating measure. For tailoring this scale to the target adolescents, based on existing recommendations [35], we selected five emotions pertinent to the prominent affective state of this age group—joy, celebration, contentment, relaxation, and success. Participants, using the same 7-point Likert scale, indicated the extent of their urge to eat in response to each positive emotion.

BMI data were acquired through survey responses, wherein students provided their body composition estimates, specifically, their weight in relation to height. The HRQoL assessment included statements from prior research [44], covering dimensions of physical, emotional, and social well-being. Participants expressed their sentiments by using a Likert-type scale, ranging from 1 (*strongly disagree*) to 7 (*strongly agree*).

## 4. Results

### 4.1. Preliminary Analyses

The survey initially invited 2749 students aged 13–17, enrolled in the ISD schools. A total of 2117 students willingly participated. However, 273 responses were deemed incomplete due to missing information, especially regarding preferences for favorite sports, specific teams, and athletes in response to an open-ended question. From the valid total of 1844 samples, the gender distribution was fairly balanced, with 53.3% identifying as male. Concerning the racial composition, the majority were White (42%), followed by Black (39%), Hispanic (11%), and others (8%). In terms of age distribution, participants were as follows: 17 years old (23%), 16 years old (18%), 15 years old (21%), 14 years old (14%), and 13 years old (17%). Additionally, 12% of them were categorized as part of the economically disadvantaged group.

Our analysis revealed that males reported higher involvement in spectator sports (M = 4.60, SD = 0.25) compared with females (M = 3.90, SD = 0.31). Economic disadvantage had a more pronounced impact on males, leading to higher levels of psychopathy (M = 3.80, SD = 0.22) compared with females (M = 3.31, SD = 0.25). Age differences were also evident, with younger adolescents (ages 13–15) experiencing higher levels of cyber victimization (M = 3.72, SD = 0.38) than older adolescents (ages 16–17, M = 2.90, SD = 0.35). Additionally, Machiavellianism was higher in males (M = 3.50, SD = 0.29) compared with females (M = 3.08, SD = 0.28), while narcissism showed a substantial difference by age, with younger adolescents (ages 13–15) reporting higher levels (M = 3.73, SD = 0.25) than older adolescents (ages 16–17, M = 3.31, SD = 0.18). Regarding emotional eating behaviors, younger adolescents showed higher positive emotions-driven eating (M = 5.20, SD = 0.31) compared with older adolescents (M = 4.82, SD = 0.26). Negative emotions-driven eating was higher in males (M = 3.41, SD = 0.19) than in females (M = 2.77, SD = 0.15). In terms of BMI, males had a higher mean (M = 22.5, SD = 3.7) compared with females (M = 20.6, SD = 3.3). HRQoL scores were higher in younger adolescents (M = 5.21), SD = 0.27) compared with older adolescents (M = 4.78, SD = 0.34). The overview of descriptive statistics and correlations are presented in Table 1.

### 4.2. Measurement Model

This study evaluated the measurement model by using a confirmatory factor analysis (CFA) that included all measures. The model yielded poor fit outcomes. Following a comprehensive examination of modification indices and loadings, specific items were identified for removal to improve the precision and validity of the survey instrument. In the Machiavellianism factor, items such as *“You should wait for the right time to get back at people”* and *“Make sure your plans benefit yourself, not others”* were deemed unsuitable and subsequently excluded. Similarly, within the narcissism factor, three reversed items reflecting aversion to attention, discomfort with compliments, and self-perception as an average person were removed for refinement. The psychopathy domain witnessed the exclusion of items related to avoiding dangerous situations, legal trouble, and a willingness to say anything to achieve one’s desires. Meanwhile, adjustments were made to the HRQoL factor, involving the removal of items associated with feelings of sadness and loneliness, as well as considerations of personal time, fair treatment by parents or guardians, and the ability to pay attention at school. These adjustments, informed by existing recommendations [45], were essential to addressing concerns regarding excessively high item correlations, causing discriminant and convergent validity problems. After removing these items, the measurement model exhibited good fit to the data (*χ*^2^ = 496.81, *df* = 180, RMSEA = 0.04, CFI = 0.92, SRMR = 0.05, *p* < 0.001). As displayed in Table 2, all average variance extracted (AVE) measures met the acceptable standard of 0.50, ranging from 0.53 (cyber victimization) to 0.77 (positive emotions-driven eating). Additionally, the discriminant validity of the measures was deemed satisfactory. The results are shown in Table 2.

### 4.3. Structural Model

In this study, a structural equation modeling analysis was employed. The initial structural model, which is shown below in Figure 1, did not converge. Consequently, the hypothesized model underwent re-specification based on both statistical considerations (covariance matrix and modification indices) and theoretical insights from the social psychology [3,46] and public health literature [30,37]. Figure 2 illustrates the modified model, which introduced four additional direct associations: causal relationships between (1) economic disadvantage and negative emotions-driven eating, (2) spectator sports involvement and positive emotions-driven eating, (3) cyber victimization and negative emotions-driven eating, and (4) narcissism and HRQoL. The modified model demonstrated an acceptable fit (*χ*^2^ = 5359.62, *df* = 1823, RMSEA = 0.04, CFI = 0.93, SRMR = 0.05, *p* < 0.001).

The path between economic disadvantage and negative emotions-driven eating links socioeconomic stressors to emotional regulation and eating behaviors [11,26]. Similarly, the connection between spectator sports involvement and positive emotions-driven eating suggests that sports enhance positive emotions and affective eating [15,27]. The link between cyber victimization and negative emotions-driven eating follows the digital self-defense hypothesis, positing that online harassment leads to emotional distress and maladaptive eating [17]. Additionally, the association between narcissism and HRQoL is supported by the literature on narcissistic traits and quality-of-life variations [21].

The specific path coefficients in the model reveal a positive association between economic disadvantage and both Machiavellianism (β = 0.33, *p* < 0.001) and psychopathy (β = 0.45, *p* < 0.001). Additionally, economic disadvantage is positively linked to negative emotions-driven eating (β = 0.34, *p* < 0.001). Spectator sports involvement emerges as positively correlated with both Machiavellianism (β = 0.67, *p* < 0.001) and narcissism (β = 0.11, *p* = 0.04), alongside a positive association with positive emotions-driven eating (β = 0.38, *p* = 0.001). Cyber victimization, on the other hand, exhibits a negative association with narcissism (β = –0.15, *p* = 0.04) while showing positive associations with both psychopathy (β = 0.21, *p* = 0.01) and negative emotions-driven eating (β = 0.46, *p* < 0.001). These results support *H1a*, *H1c*, *H2a*, *H2b*, and *H3c* while rejecting *H1b*, *H2c*, *H3a*, and *H3b*.

Machiavellianism is identified as positively related to both positive emotions-driven eating (β = 0.18, *p* = 0.02) and negative emotions-driven eating (β = 0.18, *p* = 0.01). Narcissism displays a positive connection with positive emotions-driven eating (β = 0.34, *p* < 0.001) and HRQoL (β = 0.29, *p* < 0.001), coupled with a negative association with negative emotions-driven eating (β = –0.21, *p* = 0.01). Psychopathy, meanwhile, demonstrates positive associations with both positive emotions-driven eating (β = 0.29, *p* = 0.008) and negative emotions-driven eating (β = 0.33, *p* < 0.001). Accordingly, *H4a*, *H4b*, *H4c*, *H5a*, and *H5c* are supported, but *H5b* is rejected. Additionally, positive emotions-driven eating is found to be positively associated with BMI (β = 0.34, *p* < 0.001) and HRQoL (β = 0.10, *p* = 0.04), while negative emotions-driven eating exhibits a negative association with BMI (β = –0.58, *p* < 0.001) but a positive association with HRQoL (β = 0.69, *p* < 0.001). These results indicate empirical support solely for *H6a*, rejecting *H6b*, *H7a*, and *H7b*.

## 5. Discussion

### 5.1. Theoretical Implications

This study departs from the conventional focus on positive adolescent traits by examining the unexplored domain of dark personality traits. Specifically, by identifying three determinants, our goal was to illuminate challenges faced by adolescents and contribute to the literature in this domain. Specifically, social cognitive adaptation and evolutionary perspectives [11,12,13] propose that traits such as strategic manipulation and impulsivity once conferred advantages in navigating challenges (e.g., economic hardship; [26]) but may manifest as unhealthy coping mechanisms. Our findings challenge traditional views on adolescent traits [47], confirming the role of psychological adaptations in challenging environments. 

Our findings reveal some counterintuitive results, such as cyber victimization reducing narcissism and negative emotions-driven eating lowering BMI. One possible explanation for cyber victimization reducing narcissism is that repeated online harassment may lead adolescents to become more introspective and develop a more realistic self-concept, thus reducing self-centered behaviors as a coping mechanism. This aligns with research suggesting that adverse social experiences can lead to personality changes aimed at enhancing social harmony and reducing conflict [46]. The negative association between negative emotions-driven eating and BMI could be due to adolescents choosing less calorie-dense foods or experiencing higher stress-induced metabolic rates, offsetting potential weight gain. Emotional distress might lead to reduced overall food intake due to loss of appetite, resulting in lower BMI. This is supported by studies indicating that stress and emotional turmoil can suppress appetite and lead to weight loss [38].

Our study also taps into the relationship between spectator sports engagement and dark personality traits. Traits such as strategic manipulation (Machiavellianism) and an inflated sense of self-importance (narcissism) seem drawn to the competitive ethos of sports; while past research suggests the achievement of success for individuals with these traits in athletic environments [29] we caution against potential drawbacks, including poor sportsmanship and an unhealthy fixation on winning. Our findings, therefore, challenge established narratives, empirically supporting the connection between dark personality traits and sports engagement. This contributes significantly to the literature on the psychology of physical activity, viewership, and health [29,41].

Furthermore, our exploration into the digital era uncovers a link between online threats and the adoption of psychopathic traits as a defense mechanism. The digital self-defense hypothesis [17,18,19] posits a conscious or subconscious use of psychopathic traits, such as calculated coldness and manipulation, to shield against online threats [48]. Contrary to expectations, our study revealed a shift in narcissism, as digital adversity prompts a recalibration of self-perception, reducing exaggerated self-importance. This dual impact of online threats on adolescents suggests a strategic response marked by emotional detachment [48] and manipulation [46]. The findings presented in this study defy existing notions, contributing to the literature on digital victimization [17] and offering insights into adolescents’ adaptation to digital adversity [8].

With respect to the association between dark personality and emotional eating, our study indicates that adolescents with Machiavellian tendencies strategically employ positive and negative emotions-driven eating for emotional regulation. This finding highlights the strategic nature of Machiavellian emotional regulation, extending its influence beyond typical social interactions [10]. Meanwhile, the connection between narcissism and emotional eating took an unexpected turn in our study. While intensifying positive emotions-driven eating, it unexpectedly diminishes negative emotions-driven eating. This reveals a multifaceted nature, with the complicated aspect of using celebratory meals for self-proclamation [37] and vulnerable narcissism prioritizing assertiveness over emotional eating to preserve an unblemished façade [22]. Our findings debunk simplistic narratives about narcissists and food [7], emphasizing an interplay between self-aggrandizement and emotional regulation [38].

This study sheds light on the complex relationship among emotional eating, BMI, and HRQoL. The positive link among positive emotions-driven eating, BMI, and HRQoL aligns with the conventional understanding of emotional eating’s impact on physical health [2] and its potential positive effects on psychological well-being [27]. Contrary to conventional findings, our research challenges the simplistic “negative emotions = stress eating = weight gain” model [38], revealing a negative association between negative emotions-driven eating and BMI in adolescents. This result may imply a more complex interplay influenced by individual differences and coping mechanisms. Future research should explore specific negative emotions’ varying influences on the BMI–emotional eating relationship in adolescents.

### 5.2. Practical Implications

The outcomes of this study have important implications for interventions aimed at improving well-being. Specifically, we advocate for targeted interventions for economically challenged adolescents, recognizing the potential emergence of Machiavellianism and psychopathy as adaptive responses to financial constraints. To address this, we recommend the development and implementation of focused programs [49], such as mindfulness-based stress reduction and cognitive–behavioral therapy [50], tailored to effectively address these adaptive responses. Key implementation challenges could include securing funding, providing proper training, and achieving adolescent engagement, which can be addressed through collaboration and awareness campaigns. As for economically disadvantaged students, we recommend the development and implementation of problem-solving strategies and cognitive–behavioral therapy [50] tailored to effectively address these adaptive responses. Mindfulness interventions, which may focus students on their current situations, should provide tangible support and resources to alleviate financial stress and promote resilience. Collaborative efforts with schools, community organizations, and public health agencies are essential for the successful delivery of these interventions. In light of the positive correlation between sports viewership engagement and emotions-driven eating, our implications extend to initiatives fostering healthier celebratory behaviors within the sports community. Rather than exclusively addressing emotional eating, interventions should include team games and community service projects [14], offering alternative outlets for celebration.

Addressing the digital landscape requires imperative action through comprehensive digital literacy programs [47] covering online safety, cyberbullying prevention, and responsible social media use. These programs should focus on building resilience against online threats and creating a safer online environment. Most importantly, tailored interventions addressing stress-induced eating models [35] equip adolescents with effective strategies to navigate emotional distress triggered by online harassment. Furthermore, our novel insights into the connection between online threats and the adoption of psychopathic traits highlight the need for digital literacy initiatives [31,48] prioritizing the cultivation of healthier coping mechanisms. The integration of psychological support resources into these programs is important for a holistic approach to enhancing adolescents’ digital resilience and coping mechanisms.

### 5.3. Limitations and Future Suggestions

It is important to acknowledge potential limitations. First, our study relies on self-report measures, which may introduce response bias and affect data accuracy. It is recommended to incorporate objective assessments or alternative methodologies such as the implicit association test or direct observations of teachers and parents/guardians [12,51]. Second, it would be beneficial to include a broader range of covariates to better capture the complexity of adolescents’ health outcomes. For example, examining how academic performance, psychological resilience, and family dynamics [13] interact with health outcomes would provide a more holistic understanding of adolescent well-being. Third, the cross-sectional design of this study limited our ability to make causal inferences. Longitudinal studies are needed to better understand the temporal relationships among the factors in hand. Finally, our sample is geographically limited to a single school district, which may limit the generalizability of our findings to other regions and cultural contexts.

## Figures and Tables

**Figure 1 ijerph-21-00987-f001:**
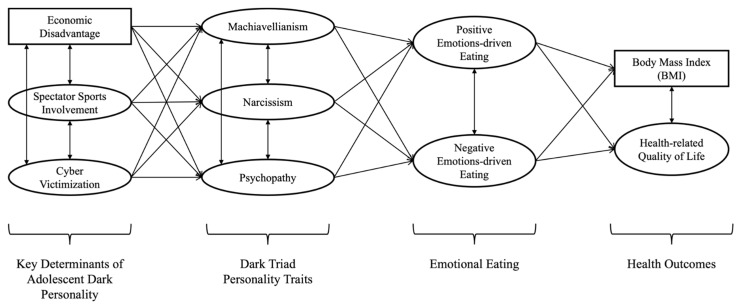
A hypothesized research model.

**Figure 2 ijerph-21-00987-f002:**
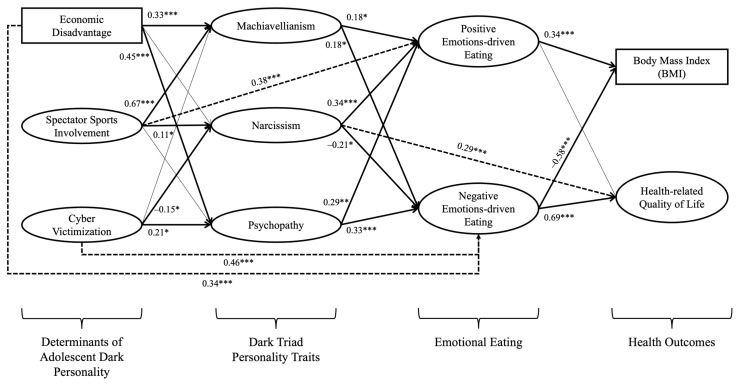
Results of the sequential mediation effects. * *p* < 0.05; ** *p* < 0.01; *** *p* < 0.001. Thick lines represent significant path coefficients, while thin lines represent non-significant path coefficients; dotted lines represent newly added direct paths. Factor correlations examined are omitted to avoid unnecessary complexity but increase readability in this figure, and the results are in Table 3.

**Table 1 ijerph-21-00987-t001:** Means (M), standard deviation (SD), and factor correlations.

Variables	M	SD	1	2	3	4	5	6	7	8	9
Economic disadvantage	0.12	0.32									
Spectator sports involvement	4.18	0.21	0.55								
Cyber victimization	3.30	0.40	0.30	0.49							
Machiavellianism	3.15	0.28	0.33	0.67	0.08						
Narcissism	3.52	0.21	0.01	0.11	–0.15	0.46					
Psychopathy	3.54	0.20	0.45	0.06	0.21	0.42	0.38				
Positive emotions-driven eating	4.95	0.22	0.34	0.38	0.46	0.18	0.34	0.29			
Negative emotions-driven eating	2.82	0.18	0.34	0.18	0.45	0.17	–0.21	0.33	0.39		
BMI	21.5	3.5	0.21	0.19	0.17	0.19	0.16	0.20	0.34	–0.58	
HRQoL	4.96	0.20	0.18	0.17	0.16	0.17	0.29	0.21	0.10	0.69	–0.25

**Table 2 ijerph-21-00987-t002:** Means (M), standard error (SE), and factor loadings (λ).

Items	*M*	*SE*	*λ*
Spectator sports involvement			
How often do you watch sports on TV, online, or attend live events?	4.18	0.21	0.73
How often do you talk with friends or family about sports?	4.37	0.19	0.68
How much do you get into watching your favorite teams and athletes’ performances or competitions?	4.11	0.19	0.77
Cyber victimization			
I have been the target of online bullying or mean messages from other kids.	3.45	0.37	0.66
I have received threatening messages or calls from other kids on my phone.	3.18	0.42	0.65
Other kids have posted hurtful or mean things about me online.	3.28	0.41	0.71
Machiavellianism			
It’s not wise to tell your secrets.	2.99	0.38	0.72
I like to use clever manipulation to get my way.	3.15	0.22	0.74
Avoid direct conflict with others because they may be useful in the future.	3.48	0.35	0.71
There are things you should hide from other people to preserve your reputation.	3.22	0.39	0.69
Narcissism			
People see me as a natural leader.	3.27	0.19	0.65
I know that I am special because everyone keeps telling me so.	3.31	0.25	0.64
I like to get acquainted with important people.	3.94	0.21	0.73
I insist on getting the respect I deserve.	3.58	0.18	0.68
Psychopathy			
I like to get revenge on authorities.	3.45	0.23	0.77
Payback needs to be quick and nasty.	4.04	0.29	0.69
It’s true that I can be mean to others.	3.48	0.16	0.74
People who mess with me always regret it.	3.19	0.22	0.78
Positive emotions-driven eating			
Joy	4.74	0.15	0.81
Celebration	5.11	0.13	0.79
Contentment	5.28	0.07	0.78
Relaxation	4.76	0.11	0.76
Success	4.84	0.11	0.75
Negative emotions-driven eating			
Sadness	2.64	0.17	0.75
Nervousness	3.22	0.12	0.72
Anger	2.57	0.18	0.67
Guilt	2.62	0.23	0.67
Not doing enough	2.85	0.15	0.64
Health-related quality of life (HRQoL)			
Felt fit and well.	4.87	0.19	0.61
Felt full of energy.	5.59	0.25	0.69
Been able to do the things that you want in your free time.	5.29	0.25	0.72
Had fun with your friends.	4.96	0.21	0.77
Felt confident at school.	4.72	0.14	0.66

**Table 3 ijerph-21-00987-t003:** The structural model with standardized path coefficients (β) and standard error (SE).

Relationship	β	SE	*p*-Value
From key determinants to dark personality traits			
Economic disadvantage → Machiavellianism	0.33 ***	0.22	***
Spectator sports involvement → Machiavellianism	0.67 ***	0.39	***
Cyber victimization → Machiavellianism	0.08	0.24	0.36
Economic disadvantage → narcissism	0.009	0.29	0.88
Spectator sports involvement → narcissism	0.11 *	0.22	0.04
Cyber victimization → narcissism	–0.15 *	0.26	0.04
Economic disadvantage → psychopathy	0.45 ***	0.29	***
Spectator sports involvement → psychopathy	0.06	0.19	0.46
Cyber victimization → psychopathy	0.21 *	0.29	0.01
From dark personality traits to emotional eating			
Machiavellianism → positive emotions-driven eating	0.18 *	0.26	0.02
Narcissism → positive emotions-driven eating	0.34 ***	0.29	***
Psychopathy → positive emotions-driven eating	0.29 **	0.34	0.008
Machiavellianism → negative emotions-driven eating	0.18 *	0.14	0.01
Narcissism → negative emotions-driven eating	–0.21 *	0.29	0.01
Psychopathy → negative emotions-driven eating	0.33 ***	0.26	***
From emotional eating to health outcomes			
Positive emotions-driven eating → BMI	0.34 ***	0.29	***
Negative emotions-driven eating → BMI	–0.58 ***	0.29	***
Positive emotions-driven eating → HRQoL	0.10	0.17	0.04
Negative emotions-driven eating → HRQoL	0.69 ***	0.19	***
Newly added direct paths			
Economic disadvantage → negative emotions-driven eating	0.34 ***	0.22	***
Cyber victimization → negative emotions-driven eating	0.46 ***	0.19	***
Spectator sports involvement → positive emotions-driven eating	0.38 **	0.41	0.001
Narcissism → HRQoL	0.29 ***	0.29	***
Within key determinants correlations			
Economic disadvantage and spectator sports involvement	0.55 ***	0.09	***
Economic disadvantage and cyber victimization	0.30 **	0.20	0.003
Spectator sports involvement and cyber victimization	0.49 ***	0.21	***
Within dark personality traits correlations			
Machiavellianism and narcissism	0.46 ***	0.22	***
Machiavellianism and psychopathy	0.42 ***	0.20	***
Narcissism and psychopathy	0.38 ***	0.09	***
Within emotional eating correlations			
Positive and negative emotions-driven eating	0.39 ***	0.18	***
Within health outcomes correlations			
BMI and HRQoL	–0.25 **	0.13	0.001

* *p* < 0.05; ** *p* < 0.01; *** *p* < 0.001.

## Data Availability

The data presented in this study are available upon request from the corresponding author due to confidentiality agreements with the school district from which the data were collected which restrict public sharing. The study involves sensitive topics, including health outcomes and personality measures. Therefore, ethical considerations prevent the open sharing of data to avoid potential harm to participants, even though no identifiable information is available in the dataset.

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
