# Peer review of "Adolescent Health and Dark Personalities: The Role of Socioeconomic Status, Sports, and Cyber Experiences"

_ijerph, 2024, doi:10.3390/ijerph21080987_

Round 1

Reviewer 1 Report

Comments and Suggestions for Authors

The paper examined how economic disadvantage, spectator sports involvement, and cyber victimization through dark personality traits conceptualized as a reaction to these phenomena affect positive and negative emotional eating, and then BMI and HRQoL. In the introductory part, I would recommend refining the presentation. While reading the introductory part, I kept asking myself why these variables were included. The hypothesized model should be announced earlier, I suggest at least at the end of the introduction, because after reading the individual hypotheses I had the constant feeling that variables related to the dark triad were chosen by chance. The method is well described, as well as the results and discussion, I especially praise the implications for practice. My perception may be shaped by the system I live in which is different from America's, but it seems a bit hypocritical to suggest mindfulness to economically disadvantaged students. If these are children who are below the poverty line, then the need to use Machiavellianism and psychopathy is logical to me as it is determined by the enormous stress that happens to them every day due to the difficulty in meeting basic needs. Using mindfulness, which will only further focus them on the current unenviable situation, could, I'm afraid, only increase stress. From my European perspective, it would be logical to propose measures by which the state ensures the satisfaction of children's needs (for example, although my country was almost at the level of third-world countries until three or four decades ago, today every child, regardless of their social status, has a free meal in school, so as not to differentiate between those who have and those who do not). I am aware that there is a different worldview in the USA, but I still think it is important to choose stress reduction strategies that are more focused on problem-solving, where the responsibility should not be on the children themselves. The paper is generally of good quality, and I hope that my comments will contribute to its quality.

Author Response

Thank you for allowing us the opportunity to revise our manuscript.  We are grateful to the reviewer for their thoughtful and constructive feedback. We have thoroughly revised our manuscript based on the reviewer’s feedback. We hope that the revised manuscript meets the standards of the International Journal of Environmental Research and Public Health. Thank you for considering our resubmission. We look forward to your feedback.

Please note that in the manuscript, newly added or revised content is indicated in YELLOW highlight for easy identification.

The paper examined how economic disadvantage, spectator sports involvement, and cyber victimization through dark personality traits conceptualized as a reaction to these phenomena affect positive and negative emotional eating, and then BMI and HRQoL. In the introductory part, I would recommend refining the presentation. While reading the introductory part, I kept asking myself why these variables were included. The hypothesized model should be announced earlier, I suggest at least at the end of the introduction, because after reading the individual hypotheses I had the constant feeling that variables related to the dark triad were chosen by chance.

  • Thank you for this feedback. We have revised the introductory section to clearly articulate the rationale for including the specific variables in our study. We have also introduced the hypothesized model at the end of the introduction to provide a coherent context for the subsequent hypotheses. We hope this addressed the concern regarding the perceived arbitrariness of the variable selection.

The method is well described, as well as the results and discussion, I especially praise the implications for practice. My perception may be shaped by the system I live in which is different from America's, but it seems a bit hypocritical to suggest mindfulness to economically disadvantaged students. If these are children who are below the poverty line, then the need to use Machiavellianism and psychopathy is logical to me as it is determined by the enormous stress that happens to them every day due to the difficulty in meeting basic needs. Using mindfulness, which will only further focus them on the current unenviable situation, could, I'm afraid, only increase stress. From my European perspective, it would be logical to propose measures by which the state ensures the satisfaction of children's needs (for example, although my country was almost at the level of third-world countries until three or four decades ago, today every child, regardless of their social status, has a free meal in school, so as not to differentiate between those who have and those who do not). I am aware that there is a different worldview in the USA, but I still think it is important to choose stress reduction strategies that are more focused on problem-solving, where the responsibility should not be on the children themselves.

  • We appreciate your insightful comments and have revised the discussion section to address these concerns. Specifically, we have incorporated a discussion on alternative stress reduction strategies that focus on problem-solving and advocate for systemic interventions to support economically disadvantaged students. We have also clarified the contextual differences and limitations in applying mindfulness universally, highlighting the need for tailored approaches based on the socio-economic context.

The paper is generally of good quality, and I hope that my comments will contribute to its quality.

  • Thank you for your positive assessment of our paper and for the valuable feedback provided. We believe the revisions have strengthened the manuscript and enhanced its clarity and relevance. Thank you again for your time and invaluable feedback.

Reviewer 2 Report

Comments and Suggestions for Authors

The authors addressed an important and under-researched topic: the impact of dark personality traits (Machiavellianism, narcissism, psychopathy) on adolescent health outcomes, with particular attention to the role of economic disadvantage, spectator involvement in sports, and cyber victimization. The study makes some noteworthy contributions, but also has areas for improvement. A detailed critique follows:

1. More details are needed on the measures used, especially for spectator sports involvement, cyber victimization, and HRQoL. Sample items, response scales, sources, and psychometric properties should be provided to allow evaluation of validity and reliability. Using established scales where possible is preferable.

2. The Results section jumps straight into CFA and SEM without descriptive statistics. Providing means, SDs, and correlations for key variables in a table would help readers understand the data. Differences by gender, age, etc. could also be insightful.

3. The rationale for some of the post-hoc model modifications, such as the added direct paths, needs more explanation. It's unclear whether these were based purely on modification indices or also had theoretical justification. Caution is warranted to avoid overfitting the model to the data.

4. The Discussion could explore deeper into the mechanisms underlying some of the counterintuitive findings (e.g., cyber victimization reducing narcissism, negative emotional eating lowering BMI). What might explain these effects psychologically or behaviorally? Offering alternative explanations and identifying areas for further research would strengthen the implications.

5. Practical implications for intervention could be more specific and actionable. What would emotionally-focused vs. behaviorally-focused programs actually entail? How can digital literacy initiatives be implemented effectively for this age group? Concrete recommendations are needed.

6. Limitations should mention the cross-sectional design, which precludes causal claims, and the geographically limited sample. Longitudinal studies and replication in other cultural contexts would bolster the generalizability of the findings.

Addressing the measurement and analysis problems mentioned above and expanding the front-end descriptions and interpretation and back-end implications would further elevate the impact of the work.

Author Response

Thank you for allowing us the opportunity to revise our manuscript.  We are grateful to the reviewer for their thoughtful and constructive feedback. We have thoroughly revised our manuscript based on the reviewer’s feedback. We hope that the revised manuscript meets the standards of the International Journal of Environmental Research and Public Health. Thank you for considering our resubmission. We look forward to your feedback.

Please note that in the manuscript, newly added or revised content is indicated in YELLOW highlight for easy identification.

The authors addressed an important and under-researched topic: the impact of dark personality traits (Machiavellianism, narcissism, psychopathy) on adolescent health outcomes, with particular attention to the role of economic disadvantage, spectator involvement in sports, and cyber victimization. The study makes some noteworthy contributions, but also has areas for improvement. A detailed critique follows:

  • Thank you for your insightful feedback on our study. We value the opportunity to address your concerns in our research. Please find below our responses to the issues highlighted.

  1. More details are needed on the measures used, especially for spectator sports involvement, cyber victimization, and HRQoL. Sample items, response scales, sources, and psychometric properties should be provided to allow evaluation of validity and reliability. Using established scales where possible is preferable.
  • We appreciate this comment. We have added detailed descriptions of the measures used for spectator sports involvement, cyber victimization, and HRQoL, including sample items, response scales, sources, and psychometric properties. We have ensured that established scales were used to enhance the validity and reliability of our findings.

  1. The Results section jumps straight into CFA and SEM without descriptive statistics. Providing means, SDs, and correlations for key variables in a table would help readers understand the data. Differences by gender, age, etc. could also be insightful.
  • We appreciate the suggestion to enhance the Results section. In response, we have added a comprehensive table presenting the means, SDs, and correlations for key variables to provide a clearer understanding of the data. Additionally, we have included an analysis of differences by gender, age, and other relevant demographic factors to offer deeper insights into the variability within our sample.

  1. The rationale for some of the post-hoc model modifications, such as the added direct paths, needs more explanation. It's unclear whether these were based purely on modification indices or also had theoretical justification. Caution is warranted to avoid overfitting the model to the data.
  • We appreciate your attention to this issue. We have added a detailed explanation for the post-hoc model modifications, clarifying that these were based not only on modification indices but also on theoretical justifications. We have taken care to avoid overfitting by ensuring that the modifications are theoretically justified and reasonable.

  1. The Discussion could explore deeper into the mechanisms underlying some of the counterintuitive findings (e.g., cyber victimization reducing narcissism, negative emotional eating lowering BMI). What might explain these effects psychologically or behaviorally? Offering alternative explanations and identifying areas for further research would strengthen the implications.
  • We appreciate your insightful suggestion. We have expanded the Discussion section to provide a more in-depth discussion of the mechanisms underlying the counterintuitive findings. We have offered psychological and behavioral explanations for these effects and identified areas for further research to strengthen the implications of our study.

  1. Practical implications for intervention could be more specific and actionable. What would emotionally-focused vs. behaviorally-focused programs actually entail? How can digital literacy initiatives be implemented effectively for this age group? Concrete recommendations are needed.
  • Great point! We have provided more specific and actionable practical implications for intervention. We have elaborated on what emotionally-focused versus behaviorally-focused programs would entail and offered concrete recommendations for implementing digital literacy initiatives effectively for adolescents.

  1. Limitations should mention the cross-sectional design, which precludes causal claims, and the geographically limited sample. Longitudinal studies and replication in other cultural contexts would bolster the generalizability of the findings.
  • We appreciate your comment once again. We have revised the Limitations section to explicitly mention the cross-sectional design and its limitation in establishing causality. We have also discussed the geographically limited sample and suggested that longitudinal studies and replication in other cultural contexts would enhance the generalizability of our findings.

Addressing the measurement and analysis problems mentioned above and expanding the front-end descriptions and interpretation and back-end implications would further elevate the impact of the work.

  • Thank you again for your thoughtful guidance and feedback and for helping us refine the quality of our manuscript.

Round 2

Reviewer 2 Report

Comments and Suggestions for Authors

I appreciate the authors' efforts to address previous feedback and improve the paper. I have a few minor suggestions to further strengthen the manuscript:

Introduction:

- The literature review is comprehensive, but consider tightening the focus slightly. Some paragraphs could be condensed to improve flow.

Discussion:

- The interpretation of results is generally strong. Consider elaborating slightly on the unexpected negative relationship between cyber victimization and narcissism.

- The practical implications are valuable. Consider adding 1-2 sentences on implementation challenges.

I commend the authors on their thorough revisions and look forward to seeing this work published.

Author Response

We appreciate your timely and valuable feedback. Following your comments, we have streamlined the last paragraph of the introduction section to improve its flow.

In the theoretical implications section, we have elaborated on the unexpected negative relationship between cyber victimization and narcissism.

We have also added a sentence addressing the implementation challenges in the practical implications section.

Thank you.